

# Proliferation of group II introns in the chloroplast genome of the green alga *Oedocladium carolinianum* (Chlorophyceae)

Jean-Simon Brouard[*], Monique Turmel[*], Christian Otis and Claude Lemieux

Département de biochimie, de microbiologie et de bio-informatique, Institut de Biologie Intégrative et des Systèmes, Université Laval, Québec, Québec, Canada

[*] These authors contributed equally to this work.

Corresponding author
Claude Lemieux,
claude.lemieux@bcm.ulaval.ca

## ABSTRACT

**Background**. The chloroplast genome sustained extensive changes in architecture during the evolution of the Chlorophyceae, a morphologically and ecologically diverse class of green algae belonging to the Chlorophyta; however, the forces driving these changes are poorly understood. The five orders recognized in the Chlorophyceae form two major clades: the CS clade consisting of the Chlamydomonadales and Sphaeropleales, and the OCC clade consisting of the Oedogoniales, Chaetophorales, and Chaetopeltidales. In the OCC clade, considerable variations in chloroplast DNA (cpDNA) structure, size, gene order, and intron content have been observed. The large inverted repeat (IR), an ancestral feature characteristic of most green plants, is present in *Oedogonium cardiacum* (Oedogoniales) but is lacking in the examined members of the Chaetophorales and Chaetopeltidales. Remarkably, the *Oedogonium* 35.5-kb IR houses genes that were putatively acquired through horizontal DNA transfer. To better understand the dynamics of chloroplast genome evolution in the Oedogoniales, we analyzed the cpDNA of a second representative of this order, *Oedocladium carolinianum*.

**Methods**. The *Oedocladium* cpDNA was sequenced and annotated. The evolutionary distances separating *Oedocladium* and *Oedogonium* cpDNAs and two other pairs of chlorophycean cpDNAs were estimated using a 61-gene data set. Phylogenetic analysis of an alignment of group IIA introns from members of the OCC clade was performed. Secondary structures and insertion sites of oedogonialean group IIA introns were analyzed.

**Results**. The 204,438-bp *Oedocladium* genome is 7.9 kb larger than the *Oedogonium* genome, but its repertoire of conserved genes is remarkably similar and gene order differs by only one reversal. Although the 23.7-kb IR is missing the putative foreign genes found in *Oedogonium*, it contains sequences coding for a putative phage or bacterial DNA primase and a hypothetical protein. Intergenic sequences are 1.5-fold longer and dispersed repeats are more abundant, but a smaller fraction of the *Oedocladium* genome is occupied by introns. Six additional group II introns are present, five of which lack ORFs and carry highly similar sequences to that of the ORF-less IIA intron shared with *Oedogonium*. Secondary structure analysis of the group IIA introns disclosed marked differences in the exon-binding sites; however, each intron showed perfect or nearly perfect base pairing interactions with its target site.

**Discussion**. Our results suggest that chloroplast genes rearrange more slowly in the Oedogoniales than in the Chaetophorales and raise questions as to what was the nature of the foreign coding sequences in the IR of the common ancestor of the Oedogoniales. They provide the first evidence for intragenomic proliferation of group IIA introns in the Viridiplantae, revealing that intron spread in the *Oedocladium* lineage likely occurred by retrohoming after sequence divergence of the exon-binding sites.

## INTRODUCTION

The Chlorophyceae is a morphologically and ecologically diverse class of green algae that belongs to the Chlorophyta, a major division of the Viridiplantae (*Leliaert et al., 2012*). Like the Chlorodendrophyceae, Pedinophyceae, Ulvophyceae and Trebouxiophyceae, this class is part of the core Chlorophyta, i.e., the rapid radiation of lineages that took place following the diversification of the prasinophyte algae. The precise relationships of the Chlorophyceae with the other core chlorophyte classes are still uncertain (*Fucikova et al., 2014*; *Lemieux, Otis & Turmel, 2014*; *Leliaert & Lopez-Bautista, 2015*; *Sun et al., 2016*; *Turmel et al., 2016a*). Five orders are currently recognized in the Chlorophyceae and as revealed by phylogenetic analyses of multiple chloroplast-encoded proteins/genes from representative taxa, they form two distinct clades: the CS clade consisting of the Chlamydomonadales (or Volvocales) and Sphaeropleales, and the OCC clade consisting of the Oedogoniales, Chaetophorales, and Chaetopeltidales (*Turmel et al., 2008*; *Brouard et al., 2010*; *Lemieux et al., 2015*; *Watanabe et al., 2016*). Within the latter clade, the Oedogoniales is sister to the Chaetophorales and Chaetopeltidales.

Comparative analysis of chloroplast DNA (cpDNA) not only clarified the evolutionary relationships among and within the main groups of the Chlorophyceae, but also highlighted the remarkable plasticity of the chloroplast genome in this class and the poor conservation of ancestral structural features inherited from the bacterial progenitor of chloroplasts. Complete cpDNA sequences of 35 chlorophycean taxa are currently available in GenBank, but a number of these genomes have yet to be described in detail. Published studies include genomes from the five major lineages of the Chlorophyceae: Oedogoniales (*Brouard et al., 2008*); Chaetophorales (*Bélanger et al., 2006*; *Brouard et al., 2011*); Chaetopeltidales (*Brouard et al., 2010*; *Watanabe et al., 2016*); Chlamydomonadales (*Maul et al., 2002*; *Smith & Lee, 2009*; *Smith et al., 2010*; *Hamaji et al., 2013*; *Smith et al., 2013*; *Vasto et al., 2015*); Sphaeropleales (*De Cambiaire et al., 2006*; *Fucikova, Lewis & Lewis, 2016a*; *Fucikova, Lewis & Lewis, 2016b*).

Chlorophycean chloroplast genomes range in size from 103 kb in the sphaeroplealean *Mychonastes homosphaera* (*Fucikova, Lewis & Lewis, 2016a*) and *Mychonastes jurisii* (*Lemieux et al., 2015*) to 521 kb in the chaetopeltidalean *Floydiella terrestris* (*Brouard et al., 2010*). With 94–99 conserved genes, the gene repertoires of chlorophycean chloroplast

genomes lack a number of protein-coding genes (e.g., *accD*, *minD*, *psaI*, *rpl19*, *tilS* and *ycf20*) compared to most other chlorophyte genomes. Genome size appears to be positively correlated with the proportion of noncoding sequences (intergenic spacer and introns) and repeated DNA elements, thus supporting the selfish-DNA hypothesis which stipulates that accumulation of noncoding DNA is caused by the proliferation of selfish elements, which in turn is limited by the harmful effects of these elements on host fitness (*Doolittle & Sapienza, 1980*; *Orgel & Crick, 1980*).

Before the emergence of the Chlorophyceae, extensive gene shuffling was accompanied by a major reorganization of the ancestral quadripartite architecture (*Brouard et al., 2008*). As reported for the Streptophyta (*Lemieux, Otis & Turmel, 2000*; *Lemieux, Otis & Turmel, 2016*), the ancestral quadripartite structure of chlorophyte cpDNA is characterized by the presence of two identical copies of a large inverted repeat (IR) separated from one another by large and small single-copy (SC) regions that are highly conserved in gene content (*Turmel et al., 1999a*; *Turmel, Otis & Lemieux, 2015*; *Turmel et al., 2016a*). While most of the investigated chlorophycean genomes have retained an IR that encodes the rRNA operon (consisting of *rrs*, *trnI*(gau), *trnA*(ugc), *rrl*, and *rrf*), partitioning of genes among the SC regions differs considerably relative to ancestral-type chlorophyte genomes and other chlorophyte genomes with a derived quadripartite structure (e.g., in ulvophycean cpDNAs: (*Pombert et al., 2005*; *Pombert, Lemieux & Turmel, 2006*). Importantly, extensive differences in gene partitioning patterns are even observed between the main lineages of the Chlorophyceae (*De Cambiaire et al., 2006*; *Brouard et al., 2008*). The IR is variable in size, gene content and intron content, ranging from 6.4 kb in the sphaeroplealean *Chromochloris zofingiensis* (*Fucikova, Lewis & Lewis, 2016b*) to 42 kb in *Chlamydomonas moewusii* (*Turmel, Bellemare & Lemieux, 1987*). In the OCC clade, the loss of the IR unites members of the Chaetophorales and Chaetopeltidales (*Bélanger et al., 2006*; *Brouard et al., 2010*; *Brouard et al., 2011*; *Watanabe et al., 2016*), while the sampled representative of the Oedogoniales (*Oedogonium cardiacum*) displays a 35.5-kb IR housing unusual coding sequences that were likely acquired through lateral transfer of mobile elements from an unknown donor (*Brouard et al., 2008*).

At the level of gene structure, several novelties arose in the Chlorophyceae, e.g., *trans*-spliced group II introns in four protein-coding genes (*rbcL*, *psaC*, *petD*, and *psaA*), substantial expansion of three separate protein-coding genes (*clpP*, *rps3* and *rps4*), and fragmentation of *rpoC1* and *rpoB* into distinct open reading frames (ORFs), which are not associated with any adjacent introns (*Brouard et al., 2010*). The *trans*-spliced group II introns in *psaC*, *petD* and *rbcL* have been found only in members of the OCC clade, whereas the *trans*-spliced introns inserted at two distinct sites within *psaA* are specific to the CS clade (*Brouard et al., 2010*; *Fucikova, Lewis & Lewis, 2016a*; *Watanabe et al., 2016*). The establishment of group II introns in chlorophycean lineages was obviously an important event in modelling the genomic landscape, however, their origin remains elusive. In contrast, introns belonging to group I show a more widespread distribution than group II introns across the Chlorophyceae and share orthologs at cognate sites in other classes of the Chlorophyta (*Brouard et al., 2008*).

The extraordinarily fluid architecture of the chlorophycean genome is thought to be the result of intramolecular and intermolecular recombination events between homologous and nonhomologous regions, with the presence of numerous dispersed repeats enhancing the frequency of recombinational exchanges (*Maul et al., 2002*; *Brouard et al., 2010*). But examination of closely related taxa is needed to better understand the dynamics of chloroplast genome evolution in each chlorophycean lineage.

The focus of the present study is on the Oedogoniales, an order of filamentous freshwater green algae that includes more than 700 species from three genera (*Bulbochaete*, *Oedocladium*, and *Oedogonium*) (*Guiry & Guiry, 2015*). We have analyzed the chloroplast genome of *Oedocladium carolinianum* and compared it with the IR-containing genome of *Oedogonium cardiacum* (*Brouard et al., 2008*). We report that the *Oedocladium* genome is remarkably similar to its *Oedogonium* counterpart at several levels but differs extensively in intron content, abundance of dispersed repeats, and putative coding sequences acquired by lateral transfer. The unexpected discovery of numerous group II introns carrying highly similar sequences in *Oedocladium* provided us with the opportunity to unravel the mechanism by which these introns colonized new genomic sites. Although group II intron proliferation to high copy number have been reported in several bacterial genomes (*Mohr, Ghanem & Lambowitz, 2010*; *Leclercq, Giraud & Cordaux, 2011*), similar observations in chloroplast genomes have been documented thus far only for euglenids (*Hallick et al., 1993*; *Pombert et al., 2012*), the unicellular red alga *Porphyridium purpureum* (*Perrineau et al., 2015*), and more recently, for ulvophycean green algae (*Turmel, Otis & Lemieux, 2016b*).

## MATERIALS AND METHODS

### Strains and culture conditions

*Oedocladium carolinianum* (UTEX LB 1686) and *Bulbochaete rectangularis* var. *hiloensis* (UTEX LB 954) were obtained from the Culture Collection of Algae at the University of Texas at Austin. Both strains were grown at 18 °C in C medium (*Andersen, 2005*) under alternating 12 h light and 12h dark periods.

### Sequencing, assembly and annotation of the chloroplast genome

An A+T rich organellar DNA fraction was obtained by CsCl-bisbenzimide isopycnic centrifugation of total cellular DNA from *Oedocladium* as previously described (*Turmel et al., 1999a*). DNA was sheared by nebulization and 2,000–4,000 bp fragments were recovered by electroelution after agarose gel electrophoresis. The fragments were end-polished with *Escherichia coli* Klenow fragment and T7 DNA polymerase and cloned into the pSMART-HCKan plasmid (Lucigen Corporation, Middleton, WI, USA). After filter hybridization of the plasmid library with the original DNA used for cloning, DNA templates from positive clones were amplified using the Illustra TempliPhi Amplification Kit (GE Healthcare, Mississauga, ON) and sequenced using Sanger chemistry by the "Plateforme d'Analyses Génomiques de l'Université Laval" (http://pag.ibis.ulaval.ca/seq/en/index.php). The DNA sequences, which were generated in these reactions using T3 and T7 primers as well as oligonucleotides complementary to internal regions of the plasmid DNA inserts, were edited and assembled using SEQUENCHER 4.7 (Gene Codes Corporation, Ann Arbor, MI,

USA). Genomic regions underrepresented in the clones analyzed were directly sequenced from polymerase chain reaction (PCR)-amplified DNA fragments using internal primers. Alternatively, PCR-amplified fragments were subcloned using the Invitrogen TOPO TA cloning kit (ThermoFisher Scientific, Mississauga, ON, CAN) before sequencing.

Genes and open reading frames (ORFs) were identified using a custom-built suite of bioinformatics tools allowing the automated execution of the following three steps: (1) ORFs were found using GETORF in EMBOSS (*Rice, Longden & Bleasby, 2000*), (2) their translated products were identified by BlastP (*Altschul et al., 1990*) searches against a local database of cpDNA-encoded proteins or the nr database at the National Center for Biotechnology Information, and (3) consecutive 100-bp segments of the genome sequence were analyzed with BlastN and BlastX (*Altschul et al., 1990*) to determine the approximate positions of RNA-coding genes, introns and exons. This pipeline produces a summary spreadsheet file listing the conserved genes and ORFs found in consecutive 100-bp segments, thus providing a quick portrait of the genome content. The precise positions of genes coding for rRNAs and tRNAs were identified using RNAmmer (*Lagesen et al., 2007*) and tRNAscan-SE (*Lowe & Eddy, 1997*), respectively. Boundaries of introns were located by manual modelling of intron secondary structures (*Michel, Umesono & Ozeki, 1989*; *Michel & Westhof, 1990*) and by comparing the sequences of intron-containing genes with those of intron-less homologs using various programs of the FASTA package (*Pearson & Lipman, 1988*).

## Analyses of dispersed repeats and gene order

Regions of the *Oedocladium* genome containing similar sequences were identified with LAST v7.1.4 (*Frith, Hamada & Horton, 2010*). To estimate the proportion of short repeated sequences, repeats with a minimal size of 30 bp were retrieved using REPFIND of REPuter v2.74 (*Kurtz et al., 2001*) with the options -f -p -l -allmax and were then masked on the genome sequence using RepeatMasker (http://www.repeatmasker.org) running under the AB-BLAST/WU-BLAST 2.0 search engine (http://blast.advbiocomp.com).

The minimal number of reversals required to convert the chloroplast gene order of *Oedocladium* to that of *Oedogonium* was estimated using GRIMM v2.01 (*Tesler, 2002*). The data set used for this pairwise comparison consisted of 93 genes and was prepared as described by *Brouard et al. (2011)*.

## Phylogenetic analyses

The deduced amino acid sequences of individual chloroplast protein-coding genes shared by nine chlorophyceans (see Table 1 for the names of taxa and accession numbers of chloroplast genome sequences) and the ulvophycean *Pseudendoclonium akinetum* (*Pombert et al., 2005*) were aligned using MUSCLE 3.7 (*Edgar, 2004*). A total of 61 genes met this criterion: *atpA, B, E, F, H, I, ccsA, cemA, clpP, ftsH, petB, D, G, L, psaA, B, C, J, psbA, B, C, D, E, F, H, I, J, K, L, M, N, T, Z, rbcL, rpl2, 5, 14, 16, 20, 23, 36, rpoA, B, C1, C2, rps2, 3, 4, 7, 8, 9, 11, 12, 14, 18, 19, tufA, ycf1, 3, 4, 12*. The amino acid alignments were converted into alignments of codons, and the poorly aligned and divergent regions in each codon alignment were removed using GBLOCKS 0.91b (*Castresana, 2000*) and the option

Brouard et al. (2016), *PeerJ*, DOI 10.7717/peerj.2627

Peer**J**

**Table 1** General features of *Oedocladium* and other chlorophycean chloroplast genomes.

| Genomic feature | OCC clade | | | | | CS clade | | | |
| | Oedogoniales | | Chaetopeltidales | Chaetophorales | | Chlamydomonadales | | | Sphaeropleales |
| | *Oedocladium* (KX507373) | *Oedogonium* (NC_011031) | *Floydiella* (NC_014346) | *Stigeoclonium* (NC_008372) | *Schizomeris* (NC_015645) | *Chlamydomonas* (NC_005353) | *Volvox* [a] (GU084820) | *Dunaliella* (NC_016732) | *Acutodesmus* (NC_008101) |
|---|---|---|---|---|---|---|---|---|---|
| **Size (bp)** | | | | | | | | | |
| Total | 204,438 | 196,547 | 521,168 | 223,902 | 182,759 | 203,827 | 461,064 | 269,044 | 161,452 |
| IR | 23,748 | 35,492 | –[b] | –[b] | –[b] | 22,211 | 15,948 | 14,409 | 12,022 |
| SC1 [c] | 98,887 | 80,363 | –[b] | –[b] | –[b] | 81,307 | 227,676 | 127,339 | 72,440 |
| SC2 [d] | 58,055 | 45,200 | –[b] | –[b] | –[b] | 78,088 | 200,100 | 112,887 | 64,968 |
| **A+T (%)** | 70.2 | 70.5 | 65.5 | 71.1 | 72.8 | 65.5 | 57.0 | 67.9 | 73.1 |
| **Conserved genes (no.)[e]** | 100 | 99 | 97 | 97 | 98 | 94 | 94 | 94 | 96 |
| **Introns** | | | | | | | | | |
| Group I (no.) | 7 | 17 | 19 | 16 | 24 | 5 | 3 | 21 | 7 |
| Group II (no.) | 10 | 4 | 7 | 5 | 9 | 2 | 6 | 2 | 2 |
| Fraction of genome (%) | 9.6 | 17.9 | 3.4 | 10.9 | 13.4 | 7.0 | 4.4 | 10.3 | 7.9 |
| **Intergenic regions** [f] | | | | | | | | | |
| Average size (bp) | 594 | 370 | 3,824 | 975 | 538 | 937 | 3,405 | 1,347 | 517 |
| Fraction of genome (%) | 34.3 | 22.6 | 77.8 | 44.4 | 31.5 | 49.2 | 75.3 | 54.1 | 34.3 |
| **Repeats**[g] | | | | | | | | | |
| Fraction of genome (%) | 11.3 | 1.3 | 49.9 | 17.8 | 1.2 | 16.8 | 45.5 | 8.6 | 3.0 |

**Notes.**

[a]Values provided for the *Volvox* genome should be considered as estimates because some intergenic regions could not be entirely sequenced.

[b]Because the *Floydiella*, *Stigeoclonium* and *Schizomeris* genomes lack an IR, only the total sizes of these genomes are given.

[c]Single-copy region with the larger size.

[d]Single-copy region with the smaller size.

[e]Conserved genes refer to free-standing coding sequences usually present in chloroplast genomes. Duplicated genes were counted only once.

[f]ORFs showing no sequence similarity with known genes were considered as intergenic sequences.

[g]Non-overlapping repeat elements were mapped on each genome with RepeatMasker using as input sequences the repeats ≥30 bp identified with REPuter.

−t = c, −b3 = 5, −b4 = 5 and −b5 = half. After concatenation of the codon alignments, a maximum likelihood tree was inferred using RAxML v8.2.6 (*Stamatakis, 2014*) and the GTR+Γ4 model of nucleotide substitutions. In these analyses, the data set was partitioned by gene, with the model applied to each partition. Confidence of branch points was estimated by fast-bootstrap analysis (*f* = *a*) with 100 replicates. Using the same approach previously used by *Brouard et al. (2011)*, we estimated evolutionary distances with CODEML in PAML 4.8 (*Yang, 2007*) from the best tree generated by RAxML. The codon analysis was run with the M1 branch model and default parameters.

To identify the relationships among *cis*-spliced group IIA introns from representatives of the OCC clade, intron sequences were aligned manually on the basis of intron secondary structures that were modelled manually using the consensus structure of IIA introns as prototype (*Michel, Umesono & Ozeki, 1989*; *Toor, Hausner & Zimmerly, 2001*), and poorly aligned and divergent regions were removed using GBLOCKS 0.91b (*Castresana, 2000*) with the parameters: −t = d −b2 = 9 −b4 = 5 −b5 = n. The resulting data set was analyzed using RAxML v8.2.6 (*Stamatakis, 2014*) and the GTR+Γ4 model. Confidence of branch points was estimated by fast-bootstrap analysis (*f* = *a*) with 1,000 replicates.

### Analysis of group IIA introns in *Bulbochaete* cpDNA

Positional homologs of six *Oedocladium* chloroplast group IIA introns were searched in *Bulbochaete* by PCR amplification of the cpDNA regions spanning these intron insertion sites. The reactions were performed using total cellular DNA and the following pairs of primers: 5′-TTCCGATTGGTCGTGGTCAACG and 5′-GATAAGCTTCACGTCCTGGAGG (*atpA*); 5′-AACCTCTTTCTTTAAGTTTCCG and 5′-TATACGCACCTGCTAATGTTGC (*atpI*); 5′-CGTTGGTCTGCTAGTATGATGG and 5′-CAGAACCTCCACGTAATAATTC (*petB*); 5′-ATCAGGTGTTTATCAATGGTGG and 5′-CTGGATTTTGAGCATAAGCTGC (*psaB*); 5′-CTGTTGTATTAAATGACCCTGG and 5′-GCTGCTACTGTTTCATATGTCC (*psbB*); 5′-AACGTTTAGGTGCTAATGTAGC and 5′-CACGACGTTCTTGCCATGGTTG (*psbC*). The resulting PCR products were analyzed by agarose gel electrophoresis and subsequent Sanger DNA sequencing using internal primers was carried out to confirm the presence of a group IIA intron of 725 bp in the *petB* insertion site (GenBank accession KX507372).

## RESULTS

### The chloroplast genome architecture is conserved in the Oedogoniales

The circular-mapping chloroplast genome of *Oedocladium* closely resembles its *Oedogonium* counterpart with respect to AT content, overall structure, gene content, and gene order (Fig. 1 and Table 1). At 204,438 bp, it is 7.9 kb larger and noticeably less gene-dense than the *Oedogonium* genome. Intergenic spacers account for 34.3% of the *Oedocladium* genome sequence compared to 22.6% for *Oedogonium*. Short dispersed repeats (≥30 bp) represent an important component of the intergenic regions, being 7-fold more abundant than in the *Oedogonium* genome (Table 1).

The *Oedocladium* chloroplast genome contains 100 genes that are usually conserved in other cpDNAs (Fig. 1). These genes encode 67 proteins, 30 tRNAs and three rRNAs.

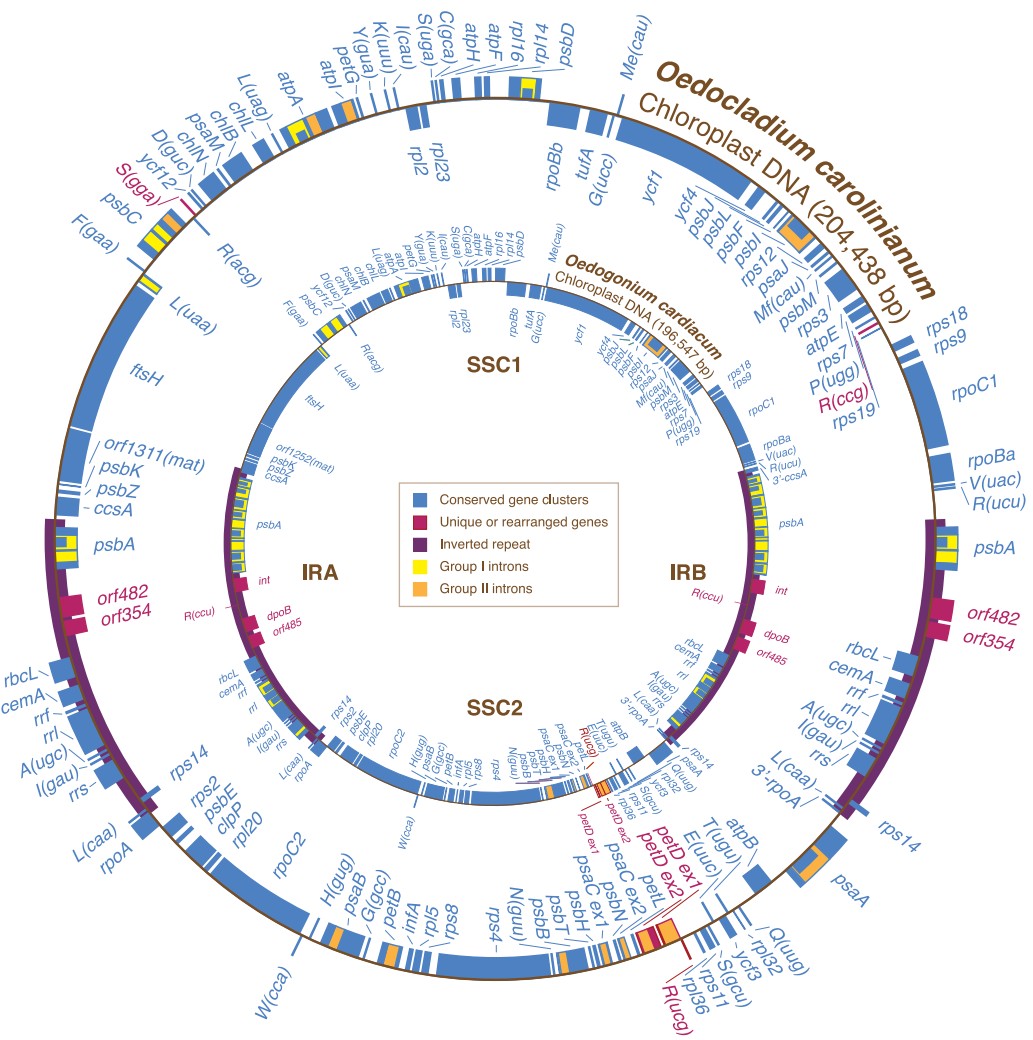

**Figure 1** **Gene maps of the *Oedocladium* and *Oedogonium* chloroplast genomes.** Coding sequences on the outside of the maps are transcribed in a clockwise direction. Genes included in blocks of colinear sequences are shown in blue, whereas rearranged or unique genes are shown in red. Group I and group II introns are shown in yellow and orange, respectively; intron ORFs are distinguished from exons by narrower boxes. The *rpoB* gene consists of two separate ORFs (*rpoBa* and *rpoBb*) that are not associated with sequences typical of group I or group II introns. tRNA genes are indicated by the one-letter amino acid code followed by the anticodon in parentheses (Me, elongator methionine; Mf, initiator methionine).

Relative to the *Oedogonium* genome, the *Oedocladium* cpDNA features two extra tRNA genes, *trnR*(ccg) and *trnS*(gga), but lacks *trnR*(ccu). In addition, *Oedocladium* shares with *Oedogonium* a large free-stranding ORF (*orf1311*) that encodes a protein with the maturase domain typically found in many group II intron-encoded proteins.

Gene order in the two oedogonialean genomes differs by a single reversal in the SC2 region (Fig. 1). The sequence that sustained this rearrangement is less than 3 kb in size and contains the *petD* and *trnR*(ucg) genes. This finding contrasts with the 20 and 24 gene rearrangements observed for pairs of closely related algae from the Chaetophorales (*Stigeoclonium*/*Schizomeris*) and the Chlamydomonadales (*Chlamydomonas*/*Volvox*)

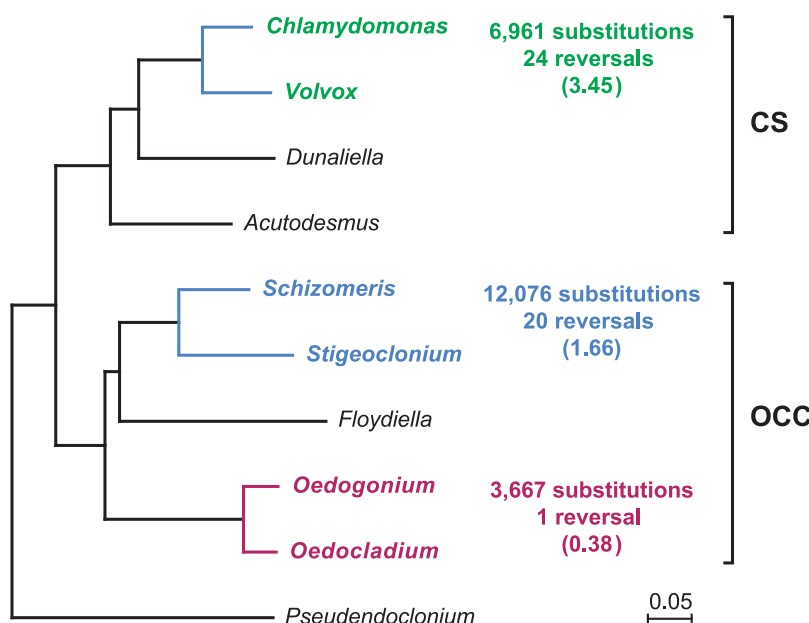

**Figure 2** **Compared evolutionary distances and chloroplast genome rearrangements in pairs of closely related green algae from the Oedogoniales, Chaetophorales, and Chlamydomonadales.** Nucleotide substitutions were estimated with CODEML (*Yang, 2007*) using the best RAxML tree inferred from a data set of 61 concatenated chloroplast protein-coding genes containing a total of 37,209 nucleotide positions. For each pairwise comparison, the minimal number of reversals required to render the genomes colinear is given together with the number of reversals per 1,000 substitutions (in parentheses). Reversals were estimated using GRIMM (*Tesler, 2002*); for the Chaetophorales and Chlamydomonadales, these estimates were taken from (*Brouard et al., 2011*).

(*Brouard et al., 2011*). To interpret these results, the evolutionary distances separating the taxa in each of these three algal pairs were estimated with CODEML using a data set of 61 concatenated protein-coding genes from nine chlorophyceans (total of 37,209 nucleotide positions) and the best RAxML tree inferred from this data set (Fig. 2). The evolutionary distance between *Oedocladium* and *Oedogonium* was found to be much smaller than those separating *Stigeoclonium* from *Schizomeris* and *Chlamydomonas* from *Volvox*. Assuming that substitutions occur at the same rate in all chlorophycean lineages, chloroplast genes rearranged at a slower pace in the Oedogoniales (0.38 reversal per 1,000 nucleotide substitutions) than in the Chaetophorales (1.66 reversals per 1,000 substitutions) and Chlamydomonadales (3.45 reversal per 1,000 substitutions) (Fig. 2).

**The *Oedocladium* and *Oedogonium* IRs display important differences**
Although the *Oedocladium* and *Oedogonium* IRs contain the same set of conserved genes (Fig. 1), they exhibit changes at several levels. First, the *Oedocladium* IR is 11.7 kb smaller, owing mainly to the absence of 11 group I introns in *psbA*, *rrs* and *rrl*. Second, the boundaries of the IR with the SC1 and SC2 regions are located at slightly different positions due to the expansion or contraction of the IR, a process known as the ebb and flow (*Goulding et al., 1996*). The *Oedogonium* IR includes 390 bp of the *ccsA* coding sequence at the junction with the SC1 region; however, the same IR/SC1 junction falls within the

*psbA/ccsA* intergenic spacer in *Oedocladium*. At the IR/SC2 junction, the *Oedocladium* IR encompasses a greater portion of the *rpoA* gene than the *Oedogonium* IR (333 bp versus 159 bp). Third, the *Oedocladium* IR is missing three unusual coding sequences that reside between *rbcL* and *psbA* in its *Oedogonium* counterpart, namely *trnR*(ccu) and the sequences coding for a protein from the tyrosine recombinase family (*int*) and a type B DNA-directed DNA polymerase (*dpoB*); instead, it features the *orf482* and *orf354*. BlastP analyses revealed that the latter ORFs display similarities with cpDNA sequences from the prasinophytes *Pyramimonas parkeae* and *Nephroselmis olivacea* and the chlamydomonadalean alga *Pleodorina starrii*. The protein product of *orf482* is similar to the putative phage DNA primases encoded by the *Pyramimonas orf454* (*E*-value = 2e − 44) (*Turmel et al., 2009*) and *Pleodorina orf571* (*E*-value = 2e − 23) (*Smith et al., 2013*), which occur within or near the IR, respectively. With regards to the *Oedocladium orf354* product, it is similar (*E*-value = 3e − 11) to the hypothetical protein encoded by the ORF (*orf594*) found next to a phage DNA primase coding sequence (*orf389*) in the *Nephroselmis* IR (*Turmel, Otis & Lemieux, 1999b*; *Turmel et al., 2009*).

### The *Oedocladium* and *Oedogonium* genomes differ in intron content

Introns represent 9.6% of the *Oedocladium* genome, whereas they account for 17.9% of the *Oedogonium* genome sequence (Table 1). With seven introns from the group I class, the *Oedocladium* genome has a deficit of ten group I introns compared with the *Oedogonium* cpDNA. All seven *Oedocladium* group I introns have homologs at the same insertion sites in previously analyzed cpDNAs from members of the OCC clade (Fig. 3). Three of these introns encode homing endonucleases from the LAGLIDADG, H-N-H, and GIY-YIG families.

The *Oedocladium* genome features six additional group II introns relative to the *Oedogonium* cpDNA, for a total of ten: six IIA and four IIB introns (Fig. 3). Note here that group IIA introns differ from group IIB introns in several regions of their secondary structures (ε′ region, length of domain ID(iv), EBS2 region, domain ID(iii)2, domain III internal loop, and linkers between domains I–VI) and in their mode of exon recognition by domain I during RNA splicing and mobility functions (*Michel & Ferat, 1995*; *Toor, Hausner & Zimmerly, 2001*; *Lambowitz & Belfort, 2015*). The *trans*-spliced IIB introns in the *Oedocladium petD* and *psaC* genes occur at the same insertion sites as similar introns found in all previously examined members of the OCC clade. With regards to the remaining *Oedocladium* introns, which are all *cis*-spliced, only the IIA and IIB introns located in *psbB* and *psbI*, respectively, share insertion sites with another member of the OCC clade, namely *Oedogonium*. Of all the group II introns examined in the OCC clade, just the *cis*-spliced IIB introns present in *psbI* and *psaA* contain ORFs (*orf476* and *orf622*, respectively); as expected, the latter sequences, which encode multifunctional reverse transcriptases, are located in domain IV of the secondary structure (Figs. S1 and S2). While the *psbI* intron-encoded protein (IEP) displays reverse transcriptase, maturase and H-N-H nuclease domains, the *psaA* IEP carries the reverse transcriptase and maturase domains but lacks the endonuclease domain.

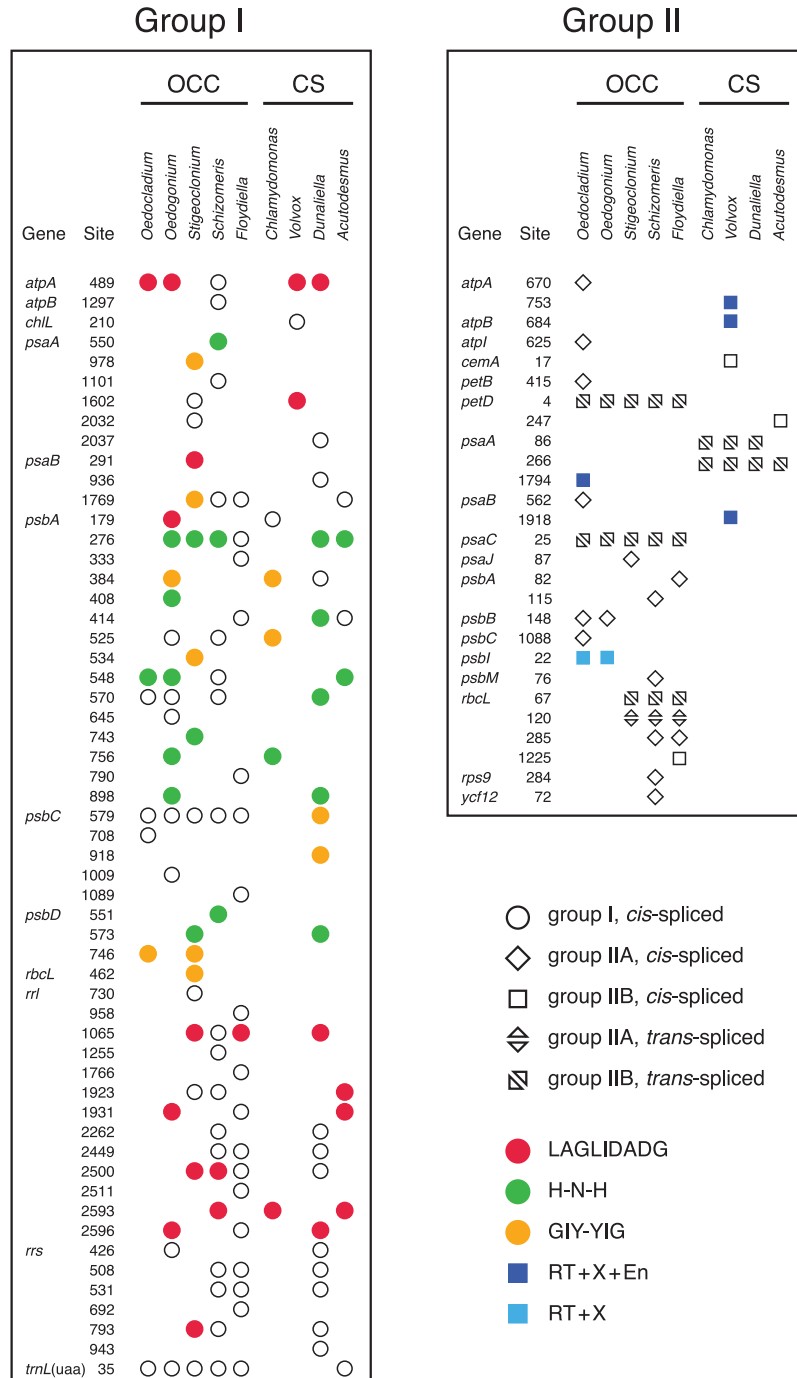

**Figure 3  Distribution of group I and group II introns in *Oedocladium* and other chlorophycean chloroplast genomes.** Open symbols indicate the absence of intron ORFs, whereas colored symbols indicate their presence (see the color code for the domain(s) encoded in the intron proteins). Intron insertion sites in protein-coding and tRNA genes are given relative to the corresponding genes in the deeply-diverging streptophyte alga Mesostigma viride (*Lemieux, Otis & Turmel, 2000*); insertion sites in rrs and rrl are given relative to *Escherichia coli* 16S and 23S rRNAs, respectively. For each insertion site, the position corresponding to the nucleotide immediately preceding the intron is reported.

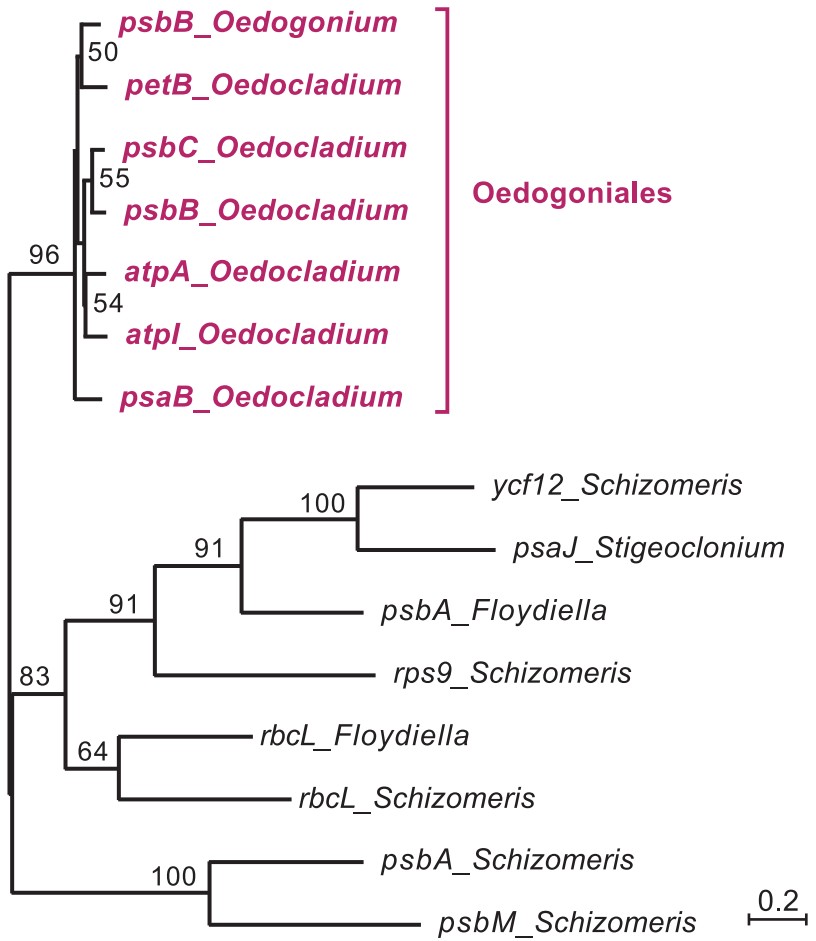

**Figure 4** **Phylogenetic relationships among the chloroplast group IIA introns of *Oedocladium* and other representatives of the OCC clade.** This tree was inferred by RAxML analysis of an alignment of 319 nucleotides corresponding to the core secondary structures of the group II introns. Bootstrap support values higher than 50% are indicated on the nodes.

## The six ORF-less group IIA introns of *Oedocladium* are closely related

In the course of identifying repeats using LAST (*Frith, Hamada & Horton, 2010*) and REPuter (*Kurtz et al., 2001*), we found that several *Oedocladium* chloroplast IIA introns display extensive regions with high levels of nucleotide identities. For instance, the *psbB* and *psbC* introns are 86.9% identical over an alignment spanning their entire lengths (693 positions). To delineate the relationships among the 15 IIA introns identified in members of the OCC clade, a global alignment of 319 nucleotides corresponding to the core secondary structures of these introns was submitted to phylogenetic analysis using RAxML under the GTR+G4 model (Fig. 4). The inferred tree features a highly robust clade (96% bootstrap support) with very short branches that contains all seven introns from the Oedogoniales. The remaining introns form separate groupings that include more divergent sequences.

### Only one *Oedocladium* group IIA intron is shared with *Bulbochaete*

We investigated whether positional homologs of the *Oedocladium* group IIA introns are present in a third representative of the Oedogoniales, *Bulbochaete rectangularis* var. *hiloensis*. For this analysis, the *Bulbochaete* cpDNA regions spanning the insertion sites of the six *Oedocladium* group IIA introns were amplified by PCR. We found that only the *petB* insertion site is occupied by an intron. Sequencing of this *Bulbochaete* intron confirmed that it belongs to the IIA subclass and that its insertion position is the same as the *Oedocladium petB* intron.

### Oedogonialean group IIA introns display high sequence variability in the elements involved in exon recognition

We compared the secondary structure models of the eight known oedogonialean IIA introns as well as their flanking 5′ and 3′ exon sequences that contain the intron-binding sites (IBS1 and IBS2) and δ′, respectively (Fig. 5). Of the 576 positions in this consensus secondary structure, 451 (78.3%) are occupied by identical residues in at least six of the eight introns. Remarkably, the three sequence motifs in domain I that play a major role in exon recognition during intron splicing and retrohoming to specific sites — i.e., the exon-binding sites EBS1 and EBS2 and δ — display high sequence variability. In spite of this variability, however, the EBS1 and EBS2 sequences of each intron show perfect or nearly perfect complementarity with the IBS sequences in the 5′ exon, and only the δ motif of the *psbC* intron cannot base pair with δ′ in the 3′ exon (Fig. 5). The alignment of the flanking exon sequences also reveals that several positions between -13 and -24 display strong nucleotide preferences, notably position -13, which displays a G in all introns.

## DISCUSSION

### Slow rate of chloroplast gene rearrangements in the Oedogoniales

Our comparison of the *Oedocladium* and *Oedogonium* chloroplast genomes disclosed high similarities not only in structure and content of standard genes but also in gene order (Fig. 1). We estimated that genes rearranged at a slower pace in the evolutionary interval separating these oedogonialean algae than in those separating *Stigeoclonium* from *Schizomeris* (Chaetophorales) and *Chlamydomonas* from *Volvox* (Chlamydomonadales) (Fig. 2). The slower rate of gene rearrangements observed for the IR-containing genomes of the Oedogoniales could possibly be explained by a stabilizing role of the IR through its participation in homologous recombination events. However, a number of studies in land plants (*Jansen & Ruhlman, 2012*; *Blazier et al., 2016*) are not consistent with the long-held hypothesis that the IR prevents gene rearrangements by limiting the frequency of recombination between single-copy regions (*Palmer, 1991*). A positive correlation has rather been observed between the level of gene rearrangements and the amount of dispersed repeats in land plant lineages displaying highly rearranged genomes (*Weng et al., 2014*; *Blazier et al., 2016*). But this correlation does not appear to hold for several green algal lineages (*Turmel, Otis & Lemieux, 2015*; *Lemieux, Otis & Turmel, 2016*), including the Oedogoniales in which the repeat-poor and gene-dense *Oedogonium* cpDNA differs from the repeat-rich *Oedocladium* cpDNA by a single reversal.

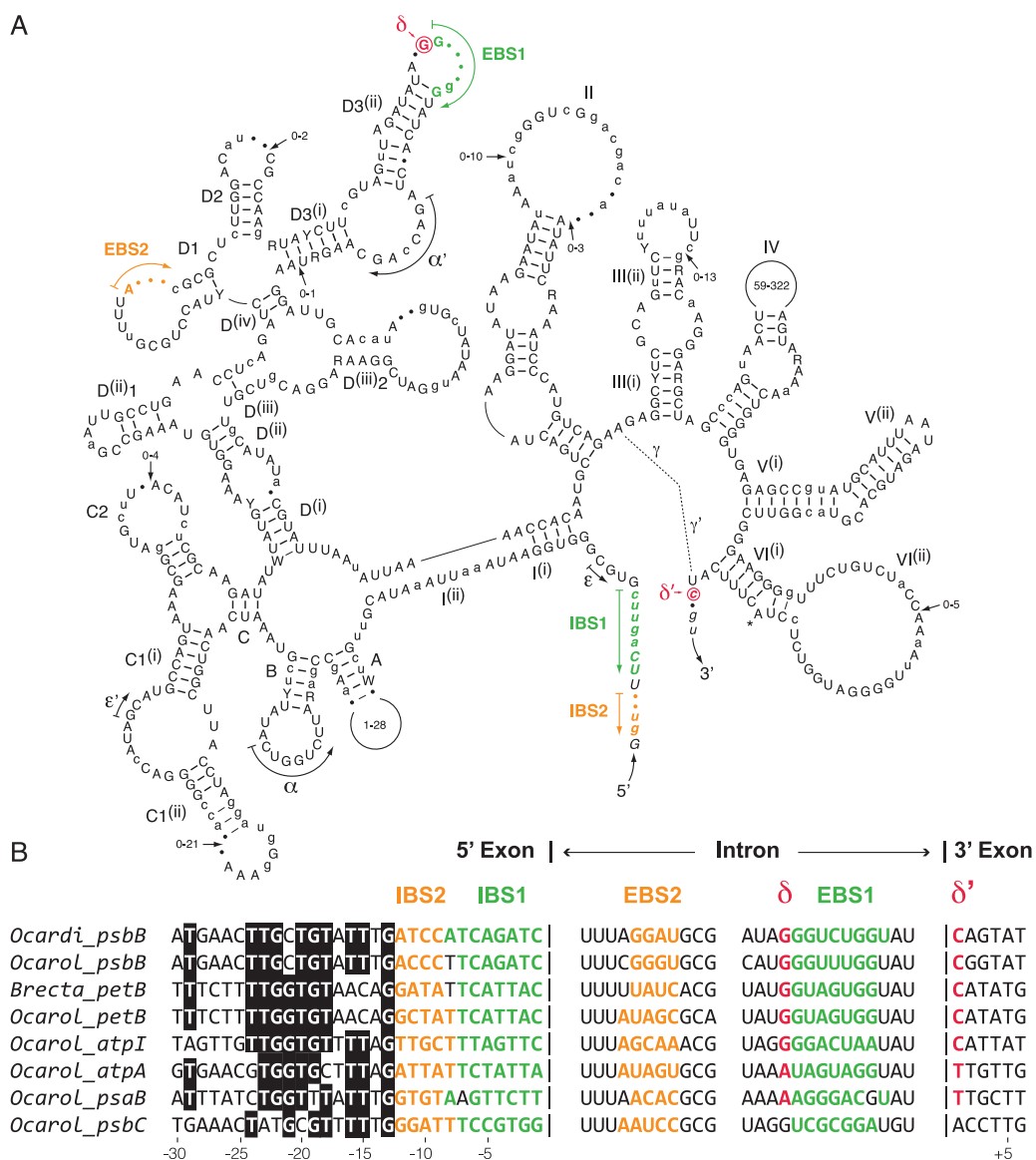

**Figure 5**  **Comparative sequence analyses of eight chloroplast group IIA introns from the Oedogoniales.** (A) Consensus intron secondary structure displayed according to *Toor, Hausner & Zimmerly (2001)*. Exon sequences are shown in italic characters. Highly conserved (in six or more introns) and slightly less conserved (in four or five introns) nucleotide positions are shown in uppercase and lowercase characters respectively; the remaining residues are denoted by dots. Highly conserved (in six or more introns) and slightly less conserved (in four or five introns) base pairings are denoted by thick and thin bars, respectively. Major structural domains and subdomains are specified by roman numerals and uppercase letters, respectively. Tertiary interactions are represented by dashed lines, curved arrows, or Greek letterings. EBS and IBS are exon-binding and intron-binding sites, respectively. The putative site of lariat formation is denoted by an asterisk. Variations in size of peripheral regions are indicated by numbers inside the loops or close to arrows. (B) Alignment presenting the exon sequences flanking the eight compared introns, along with the intron sequences containing complementary nucleotide residues (EBS1, EBS2 and δ). Colors highlight complementary nucleotide residues in the EBS/IBS and δ/ δ' sequences.

## Variable content of non-standard genes in the IR of the Oedogoniales

A surprising result that emerged from our study is the finding that the *Oedocladium* and *Oedogonium* IRs differ considerably by the nature of the freestanding, non-standard coding sequences found in the region between *rbcL* and *psbA* (Fig. 1). At this location, the *Oedogonium* IR carries a segment of about 10 kb containing sequences coding for tRNA-Arg (CCU), a type B DNA-directed DNA polymerase (*dpoB*) and a member of the tyrosine recombinase family (*int*) as well as a large ORF with no similarity to any known protein, whereas the *Oedocladium* IR displays two distinct ORFs (*orf482* and *orf354*) showing similarities to putative phage or bacterial DNA primases and to a previously reported hypothetical protein. These unusual sequences were probably gained through horizontal transfer of mobile elements, but the donor(s) remain(s) unknown. Genes with similar functions have been documented in the organelle genomes of diverse green algal lineages (*Turmel et al., 2009*; *Turmel, Otis & Lemieux, 2013*; *Civan et al., 2014*; *Leliaert & Lopez-Bautista, 2015*; *Turmel, Otis & Lemieux, 2015*; *Lemieux, Otis & Turmel, 2016*), and intriguingly, homologs of the *Oedocladium* ORFs have also been localized within or very near the IR in distantly related chlorophytes (*Turmel, Otis & Lemieux, 1999b*; *Turmel et al., 2009*; *Smith et al., 2013*). Considering this observation and the unlikely hypothesis that these sequences were transmitted by vertical descent from a very distant algal ancestor, we suggest that the IR could be a hot spot for the integration of foreign sequences. Were the unusual coding sequences in the *Oedogonium* and *Oedocladium* IRs present in the last common ancestor of the Oedogoniales or were they gained independently in these algal lineages? Chloroplast genomes from additional representatives of the Oedogoniales will need to be investigated to gain insights into these questions.

## Proliferation of group II introns in the Oedogoniales

The presence of numerous group IIA introns is a distinctive feature of the chloroplast genome in the OCC clade (Fig. 3). Of the six group IIA introns found in *Oedocladium*, four occur at unique sites, with those in *psbB* and *petB* being shared only with *Oedogonium* and *Bulbochaete*, respectively. We have demonstrated that all oedogonialean IIA introns are very closely related in sequence and that they share a common ancestry (Figs. 4 and 5). Based on this finding, our most parsimonious explanation for the observed intron distribution in the Oedogoniales is that a founding group IIA intron spread to new genomic sites during the diversification of this lineage. Although the *psbB* and *petB* introns are the most likely candidates for assuming this role, we were unable to determine with confidence the identity of the founding intron because the relationships among the oedogonialean IIA introns could not be resolved (Fig. 4) and also because the branching order of the *Oedogonium*, *Bulbochaete* and *Oedocladium* lineages remains ambiguous (*Alberghina, Vigna & Confalonieri, 2006*; *Watanabe et al., 2016*).

Remarkably, all currently known chloroplast IIA introns in the OCC clade lack ORFs (Fig. 3). Given that proteins with multiple enzymatic activities (maturases, reverse transcriptases and H-N-H endonucleases) are required to promote both the splicing and mobility of group II introns (*Lambowitz & Zimmerly, 2011*; *Lambowitz & Belfort, 2015*),

these IIA introns must rely on *trans*-acting proteins for their maintenance. The existence of a common splicing apparatus likely explains the presence of multiple ORF-less IIA introns in *Oedocladium* and perhaps the same apparatus is also used for the splicing of the other ORF-less IIA introns found in the Chaetophorales and Chaetopeltidales. In *Oedocladium*, the group II intron maturase encoded by the free-standing *orf1311* (the ortholog of *Oedogonium orf1252*) could promote splicing of the group IIA introns. We ruled out the possibility that the proteins encoded by the *psbI* and *psaA* IIB introns play this role because group IIA and group IIB introns use maturases that belong to distinct evolutionary lineages (*Toor, Hausner & Zimmerly, 2001*). Alternatively, the retention of ORF-less group IIA introns in the OCC clade may be explained by the acquisition of nuclear-encoded maturases. In *Chlamydomonas reinhartii*, it has been shown that splicing of the two *trans*-spliced group II introns in *psaA* relies on at least fourteen nuclear loci (*Merendino et al., 2006*). Although the founding intron that gave rise to the other oedogonialean IIA introns certainly encoded a protein with both maturase and reverse transcriptase activities, none of the *Oedocladium* IIA introns examined here appears to be mobile, as there is no cpDNA-encoded protein that can act in *trans* to assist their splicing and generate the ribonucleoprotein particles (RNPs) that promote mobility. Early deletion of intron ORFs might have occurred during the burst of intron proliferation as a result of an evolutionary pressure for a smaller and more compact intron structure enabling increased efficiency of splicing and mobility (*Mohr, Ghanem & Lambowitz, 2010*).

Although intragenomic proliferation of group II introns has been reported for several bacteria (*Mohr, Ghanem & Lambowitz, 2010*; *Leclercq, Giraud & Cordaux, 2011*; *Lambowitz & Belfort, 2015*), such observations are limited in the case of chloroplast genomes. In euglenids, multiple waves of group II intron acquisition undoubtedly took place, because there are remarkable differences in intron content among the chloroplast genomes of the numerous taxa that have been investigated (*Hallick et al., 1993*; *Hrda et al., 2012*; *Pombert et al., 2012*; *Bennett & Triemer, 2015*; *Dabbagh & Preisfeld, 2016*; *Kasiborski, Bennett & Linton, 2016*). The number of introns ranges from a single group II intron in the closest relative of euglenid chloroplasts, the prasinopyte *Pyramimonas parkeae* (*Turmel et al., 2009*), to over 150 introns in the *Euglena gracilis* chloroplast genome (*Hallick et al., 1993*). Although the evolutionary patterns of euglenid introns remain poorly understood, the currently available data suggest that an intron in the *psbC* gene, which encodes the reverse transcriptase/maturase *ycf13*, is the ancestral intron from which all other euglenid introns were derived and that independent waves of proliferation were associated with the acquisition of additional maturase genes (*Pombert et al., 2012*; *Bennett & Triemer, 2015*; *Kasiborski, Bennett & Linton, 2016*). Among the four classes that were sampled within the red algae, only the Porphyrididiophyceae was found to exhibit a high diversity of group II introns, with 41 intron insertion sites conserved in all five investigated strains of the unicellular *Porphyridium purpurea* in addition to the six sites displaying lineage-specific patterns of recent intron gains/losses (*Tajima et al., 2014*; *Perrineau et al., 2015*). Analyses of the small subset of introns encoding IEPs provided support for the key role played by these IEPs in intragenomic intron proliferation: indeed, most of the IEP-containing introns fell within the clade containing the lineage-specific introns (all members of the

IIB subclass) and loss of the DNA endonuclease domain was linked to a loss of mobility (*Perrineau et al., 2015*). The intron expansion reported for the Oedogoniales in the present study provides the first evidence for proliferation of group IIA introns throughout the chloroplast genome in the Viridiplantae and the second case that group II introns spread from existing introns in this phylum. In a recent report on the organelles genomes of two ulvophycean green algae from the genus *Gloeotilopsis*, dispersal of chloroplast group IIB introns was shown to have occurred independently in the Ulotrichales (*Turmel, Otis & Lemieux, 2016b*).

The studies that focused on intron dispersal from a genetic and biochemical perspective in bacteria have shown that the underlying events may differ even though, as has been shown for the well-studied *Lactococcus lactis* LI.LtrB IIA intron, all pathways involve a ribonucleoprotein (RNP) complex, which forms when the IEP binds to the intron in unspliced precursor RNA to promote splicing (*Lambowitz & Zimmerly, 2011*; *Lambowitz & Belfort, 2015*). Group II intron RNPs initiate mobility by recognizing a DNA target sequence using both the IEP and motifs within the excised intron RNA (mainly EBS sites) that base pair with the DNA target site. The intron RNA reverse splices into one strand of a duplex DNA target site (or into transiently single strand DNA at DNA replication forks) and is then reverse transcribed by the IEP to produce an intron cDNA that is integrated into the genome. Depending on the degree of base pairing interactions between group II introns and their target sites, two pathways, hereafter referred to as retrotransposition and ectopic retrohoming, have been proposed for the colonization of novel target sites (*Ichiyanagi et al., 2002*; *Mohr, Ghanem & Lambowitz, 2010*). A moderate degree of base pairing through relaxed EBS-IBS complementarity indicates that intron spread occurred through retrotransposition, the mechanism used by the LI.LtrB IIA intron (*Ichiyanagi et al., 2002*). Alternatively, introns may colonize novel sites via retrohoming following EBS sequence divergence, so that these sites become perfectly complementary with the target sites to ensure efficient splicing. The latter strategy, which has been suggested to explain the dispersal of IIB1 introns in the genome of the thermophilic cyanobacterium *Thermosynechoccus elongates* (*Mohr, Ghanem & Lambowitz, 2010*) and of a subset of group II introns in *Wolbachia* bacterial endosymbionts (*Leclercq, Giraud & Cordaux, 2011*) has the advantage of being less detrimental to the host because the introns carrying mutations promoting their insertions to new DNA target sites lose their ability to retrohome to their original homing site.

Our alignment of insertion sites for the chloroplast group IIA introns from the Oedogoniales together with the prediction of their base pairing interactions with their target sites suggest that intron dispersal was achieved by retrohoming after adaptation of the introns to new DNA target sites by selection of mutations within the EBS1, EBS2 and δ elements (Fig. 5). Indeed, in spite of the great sequence variability observed for these elements, each intron exhibits perfect or nearly perfect complementarity with its target sites in the 5′ and 3′ exons. However, we cannot eliminate the possibility that following limited divergence of the exon-binding sequences, novel genomic sites were colonized through relaxed EBS-IBS complementarity and that additional mutations were

subsequently selected to allow perfect or nearly perfect base-pairing interactions between the oedogonialean introns and their target sites. Similar analyses for the closely related chloroplast IIB introns identified in Ulotrichales (Ulvophyceae) also led to the conclusion that retrohoming played a major role in intron spread (*Turmel, Otis & Lemieux, 2016b*).

Notably, the highly conserved nucleotide residues found in the distal 5′ exon sequences of the oedogonialean IIA introns (positions −13 to −24 in Fig. 5) correspond to a region of the *Lactococcus* LI.LtrB IIA target site that is recognized by the RNP formed by the LI.LtrB intron and the intron-encoded LtrA protein (*Mohr et al., 2000*). These interactions are known to trigger initial target recognition through DNA unwinding and base-pairing of the intron RNA during reverse splicing of the intron into recipient DNA (*Lambowitz & Belfort, 2015*). The high degree of conservation observed for the distal 5′ exon sequences of the oedogonialean IIA introns suggests that this conserved region could play a similar role, thus reinforcing the hypothesis that these introns descend from the same founding intron and that intron proliferation relied on the same IEP.

## CONCLUSION

The genome data reported here for a second representative of the Oedogoniales provides deeper insights into the evolution of the chloroplast genome in the OCC clade of the Chlorophyceae. Despite remarkable similarities in structure, content of standard genes and gene order, the newly sequenced *Oedoclonium* chloroplast genome greatly differs from its *Oedogonium* counterpart at the levels of the IR-encoded genes putatively gained through horizontal DNA transfer, intron content, and abundance of dispersed repeats in intergenic regions. Compared to the IR-less genomes from the algal pair previously sampled from the Chaetophorales (*Stigeoclonium* and *Schizomeris*), genes appear to rearrange at a slower rate in the Oedogoniales, which is in line with the hypothesis that the IR may have a stabilizing role.

The widely different distribution patterns observed for the *Oedocladium* and *Oedogonium* group IIA introns is attributed to the colonization of novel genomic sites by a founding IEP-encoding intron in the *Oedocladium* lineage. Our report is the first providing evidence for intragenomic proliferation of IIA introns in the Viridiplantae. With the recent finding that IIB introns colonized novel chloroplast DNA loci in the Ulotrichales (Ulvophyceae), there are now two examples of group II intron proliferation in the Chlorophyta. In both cases, intron dispersal was likely achieved by retrohoming after adaptation of the introns to new DNA target sites by selection of mutations within the RNA sequences interacting with the exons. Given that all known oedogonialean IIA introns lack ORFs, deletion of the IEP gene probably occurred during the burst of intron proliferation to increase splicing efficiency. More chloroplast genomes will need to be sequenced for the Oedogoniales in order to identify the founding intron and also find out whether one or two separate horizontal transfer events gave rise to the foreign genes identified in the *Oedocladium* and *Oedogonium* IRs.

### Funding

This work was supported by the Natural Sciences and Engineering Research Council of Canada (http://www.nserc-crsng.gc.ca/index_eng.asp) (Grant No. 2830-2007 to MT and CL). The funders had no role in study design, data collection and analysis, decision to publish, or preparation of the manuscript.

### Grant Disclosures

The following grant information was disclosed by the authors:
Natural Sciences and Engineering Research Council of Canada: 2830-2007.

### Competing Interests

The authors declare there are no competing interests.

### Author Contributions

- Jean-Simon Brouard performed the experiments, analyzed the data, wrote the paper, prepared figures and/or tables, reviewed drafts of the paper.
- Monique Turmel conceived and designed the experiments, analyzed the data, wrote the paper, reviewed drafts of the paper.
- Christian Otis performed the experiments, analyzed the data, reviewed drafts of the paper.
- Claude Lemieux conceived and designed the experiments, analyzed the data, wrote the paper, prepared figures and/or tables, reviewed drafts of the paper.

### DNA Deposition

The following information was supplied regarding the deposition of DNA sequences:
The chloroplast genome sequences generated in this study are available from the GenBank database under the accession numbers KX507372 and KX507373.

### Data Availability

The raw data has been supplied as Supplemental File.

### Supplemental Information

Supplemental information for this article can be found online at http://dx.doi.org/10.7717/peerj.2627#supplemental-information.

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
