# Peer review of "Proliferation of group II introns in the chloroplast genome of the green alga Oedocladium carolinianum (Chlorophyceae)"

_PeerJ, doi:10.7717/peerj.2627_

## Round 0.1 · original submission · Major Revisions

Overall, the manuscript reads very well; however, I was a bit concerned that a major portion of the content appears to have been presented in a previous publication. I would pay close attention to what is actually new in this manuscript when compared to the prior publication. Is the content presented here substantially new in comparison to the prior publication? One reviewer contends that it may not. Perhaps a more detailed analysis of what is available or a more detailed comparison with the other species is called for. A rebuttal will need to include expanded reasoning on what is new here, or consider adding additional information to expand the new impacts of the manuscript. I will side with the “major revision” suggestion in that the manuscript needs to distance itself more in content with respect to the prior publication.

Reviewer 1 ·

Basic reporting

No Comments

Experimental design

No Comments

Validity of the findings

No Comments

Additional comments

The manuscript by Brouard et al. reports the proliferation group IIA introns in the Viridiplantae, and assumes that intron dispersal likely occurred by retrohoming. I have some comments listed below.

1. From the title of the manuscript, it said “Proliferation of group II introns…..”, but in the main text, the authors mainly investigated the proliferation of group IIA introns. So what are the characteristics of group IIB introns of Oedogonium?
2. On page 8, line 140: Why choosing “consecutive 100-bp segments” to identify gene sequences?
3. On page 9, line 175: Please provide the details about using “intron secondary structures” to align the intron sequences, such as the programs, paramaters.
4. On page 11, line 216: To calculate the evolutionary distance between two species (e.g. Oedogonium, Oedocladium), the authors simply sum up the number of nonsynonymous and synonymous nucleotide substitutions of both species as the distance. I am not sure whether it is the valid measurement. Could the authors provide some references to support it?
5. On page 17, line 344: There are two “Oedogonium”, please correct it.

Reviewer 2 ·

Basic reporting

Multiple references not organized chronologically. Introduction is VERY general and lacks details and structure. Authors cite 13 papers after one single sentence. I would like to see this part of introduction developed better. I also would advise not to use words like “bloated, boast etc.”. Section on IR (and SC) seems to be critical to proper understanding of publication and should consist more details as separate paragraph/s.

Experimental design

No comments.

Validity of the findings

I found very similar paper by the same four authors from 2008 (Chloroplast DNA sequence of the green alga Oedogonium cardiacum (Chlorophyceae): Unique genome architecture, derived characters shared with the Chaetophorales and novel genes acquired through horizontal transfer. BMC Genomics). Both papers have very similar methods, analysis and results. Please compare both Figure 1 (both papers), Table 1 (both papers) , Figure 3 (both papers) and Figure 4 (in 2008 paper) vs. Figure 5 (in current submission). I understand that discussion of both paper focuses in slightly different aspects of genome evolution of mentioned species however I do not think that presented data significantly improves body of knowledge. Main difference is addition of two more genomic sequences and focus of paper toward group II introns (also mentioned in paper from 2008).

Also authors claim in current paper that: "Our report is the first providing evidence for intergenomic proliferation of IIA introns in the Viridiplantae", where in paper from 2008 they state: "The intergenic spacers account for 22.6% of the total genome sequence and vary from 22 to 1721 bp, for an average size of 370 bp. This is the lowest proportion and smallest average size of intergenic spacers observed thus far for the chloroplast genome of a photosynthetic chlorophyte belonging to the Ulvophyceae, Trebouxiophyceae or Chlorophyceae (UTC)"

In my opinion conclusion about intergenomic/intergenomic proliferation of IIA introns in Viridiplantae could be made based on data provided in Paper from 2008.

---

## Round 0.2 · accepted · Accept

Your latest revision addressed most succinctly the criticisms from the reviewers and helped to address some of the earlier concerns. This version of the manuscript stands well as a reflection on the updated assessment of genome sequence with respect to new and known data. You have provided a good case for your attempt to decipher and add hypotheses for the colonization of introns into new genomic sites. I agree that your conclusions are solid and contribute to the advancement of the research area. I hope this will suit the timing of other publications in the works. Congratulations on a fine manuscript which will be moved forward in the publication process.